# Implications of Sphingolipid Metabolites in Kidney Diseases

**DOI:** 10.3390/ijms23084244

**Published:** 2022-04-11

**Authors:** Shamroop kumar Mallela, Sandra Merscher, Alessia Fornoni

**Affiliations:** 1Katz Family Division of Nephrology and Hypertension, Department of Medicine, Miller School of Medicine, University of Miami, Miami, FL 33136, USA; sxm1446@med.miami.edu; 2Peggy and Harold Katz Family Drug Discovery Center, Miller School of Medicine, University of Miami, Miami, FL 33136, USA

**Keywords:** sphingolipids, glomerular diseases, tubulointerstitial diseases

## Abstract

Sphingolipids, which act as a bioactive signaling molecules, are involved in several cellular processes such as cell survival, proliferation, migration and apoptosis. An imbalance in the levels of sphingolipids can be lethal to cells. Abnormalities in the levels of sphingolipids are associated with several human diseases including kidney diseases. Several studies demonstrate that sphingolipids play an important role in maintaining proper renal function. Sphingolipids can alter the glomerular filtration barrier by affecting the functioning of podocytes, which are key cellular components of the glomerular filtration barrier. This review summarizes the studies in our understanding of the regulation of sphingolipid signaling in kidney diseases, especially in glomerular and tubulointerstitial diseases, and the potential to target sphingolipid pathways in developing therapeutics for the treatment of renal diseases.

## 1. Introduction

Sphingolipids (SLs) are lipids with complex structures which were named after their sphinx-like structure by Thudichium in 1870 [1]. Lipids are major components of membranes in all eukaryotic cells determining the structural and functional integrity of cells. SLs are one of the important structural components of plasma membranes in all eukaryotes, but they also act as bioactive signaling molecules with numerous cellular physiological functions that include cell adhesion, cell proliferation, cell migration, inflammatory response and apoptosis [2]. Among all SLs, ceramides, ceramide-1-phosphate (C1P), sphingosine and sphingosine-1-phosphate (S1P) are the best-known bioactive SLs. Ceramides and S1P are known to have opposing functions in the cell, thereby functioning as a sphingolipid rheostat. Ceramides regulate the pathways that lead to apoptosis, cell cycle arrest and senescence [3] whereas S1P was shown to be anti-apoptotic and promote cell survival and cell invasion [4]. Thus, a proper balance of both SLs is essential for cell survival, and dysregulation of this sphingolipid rheostat can lead to cell death [5,6]. The sphingolipid rheostat seems to play a major role in kidney diseases as there is evidence showing that an increase in ceramide levels causes podocyte damage through mitochondrial dysfunction leading to renal failure in diabetic mice [7,8,9,10]. On the other hand, S1P was shown to be essential for normal podocyte function [11]. In support of this, radiation treatment is associated with reduced levels of S1P and podocyte injury, while treatment of irradiated podocytes with S1P was shown to protect them from radiation-induced damage by attenuating cytoskeleton remodeling [12]. Though SLs are known to contribute to several diseases, including insulin resistance, inflammation, neurodegradation, cancer, and lysosomal diseases [13,14,15], in this review we will focus on the contribution of SLs to the pathogenesis and progression of kidney diseases, especially those that are related to podocyte injury and tubular damage.

## 2. Sphingolipid Biosynthesis and Catabolism

SLs are composed of a sphingosine backbone linked to a fatty acid with an amide bond. Based on the difference in the hydrophilic attachments (functional groups or substituents), SLs are categorized into ceramides, sphingomyelin and glycosphingolipids (Figure 1). Ceramide is the central metabolite or substrate for the synthesis of complex SLs in the SL pathway. Ceramides can be synthesized by a de novo pathway and a salvage pathway (Figure 2).

### 2.1. De Novo Pathway

As shown in Figure 2, the first step is the condensation of L-serine with palmitoyl coenzyme-A (C16-CoA) to 3-ketosphinganine, which is catalyzed by serine palmitoyl transferase (SPT). This step is the rate-limiting step of the SL pathway. The SPT enzyme complex consists of two catalytic subunits (SPTLC1, SPTLC2) or a third regulatory subunit (SPTLC3) instead of SPTLC2. Orosomucoid-like proteins (ORMDLs) play an important role in regulating the SL biosynthesis by inhibiting the activity of the SPT complex [16]. 3-Ketosphinganine is then reduced to sphinganine or dihydrosphingosine by 3-ketosphinganine reductase (KDSR) [17]. Sphinganine can be converted to dihydroceramide by the attachment of an acyl group by dihydroceramide synthase or ceramide synthases (CerS) [18] or to phytosphingosine by the addition of a hydroxyl group at position C4 by dihyrosphingosine hydroxylase, or can be phosphorylated to form sphinganine-1-phosphate by sphingosine kinases [19]. There are six different CerS identified with a varying preference for different acyl chain lengths for the synthesis of dihydroceramide from sphinganine. For example, CerS1 preferentially uses C18; CerS2, CerS3 and CerS4 use C22, 24; C26; C18, 20, 22, respectively; whereas CerS5 and CerS6 exhibit a preference towards C14, C16 fatty acyl-CoA [20] (Figure 3). Dihydroceramide desaturase catalyzes the formation of ceramide from dihydroceramide by the addition of a 4,5 trans double bond [21] (Figure 2).

### 2.2. The Salvage Pathway

Though glycosphingolipids contribute to the synthesis of ceramides via the salvage pathway, sphingomyelinases (SMases) play a major role in this pathway by producing ceramides and phosphatidylcholine from the hydrolysis of sphingomyelin (SM) [22,23] (Figure 2). Based on their optimal pH activity, SMases are categorized as acidic (SMPD1), neutral (SMPD2, SMPD3, SMPD4 and SMPD5) and alkaline (ENPP7) SMases.

### 2.3. Catabolism of Ceramides and Synthesis of Other Sphingolipids

Ceramides are deacylated to generate sphingosine and free fatty acyl-CoA by ceramidases (CDases, Figure 2). There are three types of CDases depending on their optimal pH activity. Acid ceramidase (N-Acylsphingosine Amidohydrolase 1-ASAH1) primarily catalyzes the hydrolysis of ceramides with a chain length of C6-C18 at lysosomes. Neutral ceramidase (ASAH2) is predominantly localized at plasma membrane and can act on ceramides and dihydroceramides with a chain length of C16 and C18. Alkaline ceramidases (ACER1, ACER2, and ACER3) are primarily found at ER and in the Golgi apparatus where they break down ceramides of chain lengths C20–C24 [24].

Sphingosine generated by CDases can be transported to the ER and can be used in the synthesis of ceramides [25,26]. Sphingosine can be phosphorylated to sphingosine-1-phosphate (S1P) by two kinases, sphingosine kinase 1 and sphingosine kinase 2 [27]. At plasma membrane, S1P can be dephosphorylated by lipid phosphate phosphatases (LPP1-3) and cytosolic S1P can be dephosphorylated by S1P-specific phosphatases (SPP1 and SPP2) [26,28]. S1P can irreversibly be degraded to hexadecenal and phosphatidylethanolamine by sphingosine-1-phosphate lyase (SGPL1) [22,29,30] (Figure 2).

Ceramides can be phosphorylated to ceramide-1-phosphate (C1P) by ceramide kinase (CERK) in the Golgi apparatus. C1P can be dephosphorylated to ceramides by sphingomyelin phosphodiesterase acid like 3b (SMPDL3b) in vitro [29,30]. Ceramides can also be converted to sphingomyelin by the transfer of phosphatidylcholine to ceramide by sphingomyelin synthases (SMS) [22] (Figure 2). Ceramides can be converted to glucosylceramide by glucosylceramide synthase and to galactosylceramide by galactosylceramide synthase by adding glucose and galactose, respectively [22]. Lactosylceramide is formed by the addition of galactose to glucosylceramide by lactosylceramide synthase that can be converted to gangliosides, which plays an important role in cell–cell recognition and cell adhesion [31].

## 3. Sphingolipids in Kidney Diseases

The glomerulus of the kidney is an important structure which is essential to maintain the glomerular filtration barrier (GFB) that prevents leakage of proteins into the urine. Proper maintenance of this GFB is essential throughout life to prevent kidney injury and progression to kidney disease. Hence, understanding the factors contributing to kidney damage is important. In this review, we will focus on kidney diseases that are known to be caused due to imbalance in the levels of SLs and their association with podocyte injury and tubular damage.

### 3.1. Role of Sphingolipids in Podocyte Injury in Glomerular Diseases

The glomerular filtration barrier (GFB) is formed by podocytes, the glomerular basement membrane (GBM) and glomerular endothelial cells. Podocytes are terminally differentiated epithelial cells in the glomerulus. Foot processes of podocytes interdigitate with foot processes from adjacent podocytes, leaving filtration slits in between them, which are bridged by a structure called the slit diaphragm. Podocyte integrity, which mainly depends on the integrity of their actin cytoskeleton, is crucial for maintaining the proper function of the GFB [32]. Podocyte dysfunction and loss of podocytes from the GFB due to cellular stress, genetic mutations, inflammation or lipotoxicity can lead to the leakage of proteins into urine, referred to as proteinuria. Podocyte injury and associated proteinuria is a hallmark of glomerular diseases [33]. Podocyte injury contributes to the progression of several glomerular diseases, including diabetic kidney disease (DKD), focal segmental glomerulosclerosis (FSGS), Alport syndrome, IgA nephropathy and lupus nephritis (LN). In this section, we will focus on the glomerular diseases that are associated with imbalance in the levels of SLs and podocyte damage.

#### 3.1.1. Diabetic Kidney Disease

Diabetic kidney disease (DKD) has been tightly linked to podocyte dysfunction and the progression to end-stage kidney disease (ESKD) [34]. Loss of podocytes in diabetic patients with DKD [35,36] and alterations in the SL composition of podocytes were shown to contribute to the pathogenesis and progression of DKD [37]. Dysregulation of SL metabolism in podocytes was found to disrupt proper functioning of podocytes, thereby subsequently compromising the proper functioning of the GFB [37] leading to proteinuria, glomerular fibrosis and glomerulosclerosis. The contribution of various SLs to the progression of DKD is discussed below. 

##### Ceramides

Ceramides play a major role in DKD. In the plasma of diabetic patients, lipidomic analyses demonstrate that sphingosine, ceramide and glycosphingolipids levels were found to be increased [38,39,40], indicating that SL metabolism is dysregulated in these patients. Though it is unclear if CERS2 function influences albuminuria, it was shown that the rs267734 variant of CERS2 is associated with increased albuminuria in patients with diabetes [7]. In patients with diabetes, plasma levels of sphingosine, sphinganine, ceramides and glycosphingolipids are increased and may contribute to disease progression [38,39,40]. Non-targeted serum lipidomic analysis revealed that higher levels of sphingomyelins in the serum correlated with a lower risk of end stage renal disease (ESRD) in patients with type 1 diabetes (T1D) [41]. High coverage targeted lipidomics in a population-based cohort study in China reported that SLs, especially ceramides, sphingomyelins and S1P were positively associated with the incidence of type 2 diabetes (T2D) [42,43]. Similarly, another clinical study using large-scale lipidomics identified an association between plasma SLs, especially ceramides, sphingomyelin and S1P, and the incidence of T2D and higher concentrations of sphingomyelins were associated with a higher risk of T2D [43]. Dysregulated ceramide levels, especially an increase in the levels of long chain ceramides (C14-C20) and decreased levels of very long chain ceramides (C24), were observed in kidney cortices of diabetic *db*/*db* mice [44]. Correlation analysis identified an inverse relationship of ceramide (C16 and C24) levels in plasma and kidney cortex [44]. Increased ceramide levels can lead to inhibition of insulin-stimulated GLUT redistribution from intracellular stores to the plasma membrane by blocking the activation of AKT/PKB, a serine/threonine kinase that plays an important role in insulin-dependent physiological events [8], ultimately leading to altered kidney function. A clinical study, using targeted liquid chromatography-tandem mass spectrometry (LC-MS) analysis demonstrated that C16 and C18 ceramides are elevated in the plasma of T2D patients [45] and in the urine of patients with DKD, where C16 and C18 ceramides correlated with urinary biomarkers such as albumin [46]. Meanwhile, another study showed that lower plasma levels of very long chain ceramide species (C20–C26) were associated with the progression of proteinuria in DKD [47]. Furthermore, an inverse correlation between proteinuria and kidney tissue ceramide (C14, 16, 18, 24) levels was identified in patients with DKD during disease progression [44]. Our studies demonstrated a decrease in the total level of ceramides in kidney cortex of *db*/*db* mice [29,48], while an increase in ceramides was observed in podocytes in radiation-induced podocytopathy [12]. Podocyte-specific deletion of acid ceramidase (*Asah1*) leads to an increase in ceramide levels in glomeruli of mice along with functional and morphological changes indicating the development of nephrotic syndrome. Interestingly, acid sphingomyelinase (ASMase)-mediated increases in ceramide were found to be associated with podocyte injury and glomerulosclerosis during hyperhomocysteinemia and obesity [10], indicating that not only ceramides synthesized via the de novo pathway are toxic to the cells but ceramides produced by the action of ceramidases or sphingomyelinases are equally toxic to cells. ASAH1 plays an important role in podocyte injury and glomerulosclerosis during hyperhomocysteinemia by promoting the NADPH oxidase associated oxidative stress [10]. LC-MS analysis of urinary sphingolipids of stage 3 DKD patients showed an increase in the levels of ceramides (Cer d18:1/16:0, Cer d18:1/18:0, Cer d18:1/20:0, Cer d18:1/22:0 and Cer d18:1/24:0) in urine [46], which correlated with proteinuria and the severity of DKD [46], whereas a decrease in plasma levels of very long chain ceramides (C20–C26) in patients with T1D was shown to correlate with the progression of macroalbuminuria [47].

##### Sphingosine-1-Phosphate

S1P, which promotes cell growth and cell survival, is involved in maintaining normal podocyte function [11]. Exogenous S1P administration was shown to protect podocytes from radiation-induced damage through relocation of the actin binding protein, ezrin (which stabilizes the association of actin filaments with the plasma membrane), from the cytosol to the plasma membrane and remodeling of the actin cytoskeleton [12]. In support, SPHK1 was shown to protect podocytes from fibrosis induced by TGF-β treatment by negatively regulating the expression of connective tissue growth factor (CTGF), which plays an important role in renal fibrosis [49]. Additionally, SPHK1 protein expression and S1P levels are increased in podocytes of DKD patients compared to healthy individuals [49,50] and SPHK1 protein expression was shown to be increased in a streptozotocin (STZ)-induced mouse model of DKD [49,51] and in glomeruli of mice with Alloxan (a toxic glucose analogue that destroys insulin producing cells in the pancreas) induced diabetes in association with S1P accumulation [50]. Increase in S1P levels were also observed in STZ-treated rat glomeruli [51], which was prevented by intraperitoneal insulin injections [52], indicating a connection between insulin signaling, glucose metabolism and SPHK expression. Although the mechanism by which Apolipoprotein M (APOM) in the kidney contributes to the renal S1P content remains unclear, it is hypothesized that insulin negatively regulates APOM function (probably by increasing the susceptibility to glycosylation), thereby regulating S1P levels [52]. Interestingly, loss of sphingosine kinase 2 (SPHK2) was shown to protect podocytes from STZ-induced podocytopathy and albuminuria in mice by regulating the transcription factor Wilm’s tumor suppressor gene 1 (WT1) and enhancing the expression of the WT1 target gene Nephrin, an important component of slit diaphragm [53], which implies that STZ-induced S1P accumulation is mediated by SPHK2. Diabetes was shown to be significantly associated with a decrease in the serum levels of S1P, sphingomyelin, sphinganine and sphingosine when compared to healthy individuals [54]. LC-MS analysis of blood samples of T2D patients showed that S1P correlated with the progression of T2D and was a predictor for cardiovascular complications [54]. 

There are five S1P receptors (S1PR1–S1PR5) which mediate S1P signaling in cells. All S1P receptors’ mRNA were found in mouse kidney extracts except S1PR5 mRNA [55]. S1PR4 was shown to be expressed in immortalized mouse podocytes [56] but not in human podocytes [53]. Increased S1PR2 and decreased S1PR1 mRNA expression in mesangial cells of STZ-induced diabetic rats was shown to contribute to the progression of DKD [57]. In support of a role for S1PR signaling in kidney diseases, inhibition of S1PR2 prevented renal injury in mouse models of DKD [56], while S1PR1 activation using agonists such as SEW2871 and FTY720 prevented STZ-induced renal damage in diabetic mice by restoring the protein expression of nephrin and podocin [56]. Similarly, S1PR1 agonism was also found to reduce ischemia reperfusion injury in mouse kidneys [55]. Taken together, these studies suggest that S1P signaling mediated by S1PR2 is toxic to cells whereas S1PR1-mediated S1P signaling prevents cell damage. In addition to STZ-induced renal damage, FTY720 was also shown to attenuate the angiotensin II-induced podocyte damage by downregulating inflammatory cytokines such as tumor necrosis factor (TNF-α) and interleukin 6 (IL-6) [58]. Interestingly, mice with podocyte-specific deletion of sphingosine-1-phosphate lyase (SGPL1) develop proteinuria, which is a consequence of accumulation of S1P [59]. Importantly, a genetic mutation in SGPL1 was found in patients with nephrotic syndrome [60,61], and mice with SGPL1 deficiency were shown to exhibit severe podocyte injury and proteinuria [59]. 

##### Ceramide-1-Phosphate

Compared to ceramide and S1P, less is known about the role of ceramide-1-phosphate (C1P) and C1P signaling in kidney diseases. It was previously shown that knockout of CERK in mice prevents glomerular disease [62]. We have shown that increased SMPDL3b protein expression is observed in glomeruli of DKD patients [48], and is associated with altered SL metabolism, leading to C1P deficiency in podocytes resulting in podocyte damage [29]. We further showed that the increased SMPDL3b protein expression and C1P deficiency led to modulated insulin signaling due to impaired AKT phosphorylation [29], thereby also blocking the access of CERK to ceramide [30], a phenotype that was reversed by exogenous administration of C1P in diabetic mice [29]. From these studies, it becomes clear that C1P deficiency contributes to the progression of DKD and that targeting C1P may represent a novel option to develop treatment strategies for patients with DKD. 

##### Glycosphingolipids

Glucosylceramide levels were shown to be increased in kidneys of both *ob/ob* and STZ-induced diabetic rats [63] and lactosylceramides in diabetic *db*/*db* mice [64]. Lipidomic studies of plasma of patients with T1D revealed that the decrease in long and very long lactosylceramides was significantly associated with an increased risk of the patients to develop macroalbuminuria [65], whereas glycosphingolipids were shown to be associated with DKD and microalbuminuria in T1D patients [65]. While several classes of glycosphingolipids exist, gangliosides are known to be involved in DKD [66]. Gangliosides are glycosphingolipids that have sialic acid side chains. Gangliosides are components of cell membranes, particularly of lipid rafts, and were shown to modulate signal transduction. Uridine diphospho-N-acetylglucosamine 2-epimerase and N-acetylmannosamine kinase are the two enzymes catalyzing the sialyation of gangliosides, which play an important role in maintaining glomerular function [67]. Increase in monosialodihexosyl (GM3) gangliosides were observed in kidney cortices of diabetic rats in early stages of DKD [66] and in patients with DKD [63,68], suggesting a possible connection between GM3 and DKD. It has been suggested that GM3 contributes to the pathogenesis of DKD [63] by inactivating vascular endothelial growth factor (VEGF) receptor and interfering with AKT signaling [69]. In patients with DKD, serum sialic acid-gangliosides levels positively correlated with blood glucose, hemoglobin A1c, serum urea, serum creatinine and microalbuminuria [70]. Globotriaosylceramide (Gb3) accumulation in organelles such as lysosomes, the ER and nucleus of podocytes has been described in Fabry disease, and was found to be associated with glomerular hypertrophy and with glomerular mesangial cell widening [71]. Lyso-Gb3 was shown to cause podocyte loss and foot process effacement leading to albuminuria due to receptor interacting protein kinase 3 (RIPK3)-induced cytoskeleton remodeling and ROS generation [72]. 

All these studies indicate that glycosphingolipids are important for the proper function of podocytes and to prevent the progression of DKD.

#### 3.1.2. Focal Segmental Glomerulosclerosis 

Focal segmental glomerulosclerosis (FSGS) is a rare kidney disease that is characterized by proteinuria and podocyte injury. Like DKD, FSGS has been tightly linked to podocyte dysfunction and can lead to ESKD [34]. FSGS is the most common cause of nephrotic syndrome and glomerular diseases in adults that result in ESKD [73]. When compared to DKD, the contribution of SLs to the pathogenesis of FSGS is less studied. Deletion of SGPL1 in mice and humans causes an increase in S1P and ceramides in tissues, resulting in steroid-resistant nephrotic syndrome with mesangial hypercellularity, glomerular hypertrophy, glomerular fibrosis and diffused mesangial sclerosis [61,62,74,75]. In mice, partial deficiency or inhibition of SGPL1 leads to podocyte foot process effacement and proteinuria [59]. Similarly, podocyte-specific deletion of *Asah1* was shown to induce podocytopathy and nephrotic syndrome in mice due to ceramide-induced oxidative stress followed by apoptosis in glomeruli [76] and lysosomal dysfunction [77], while lysosomal deficiency of Asah1 led to the development of membranous nephropathy, FSGS and minimal change disease [74]. Therefore, targeting the SL pathway could also represent a potential therapeutic strategy to delay, prevent or treat podocyte injury in patients with FSGS. Analysis of dried blood spots of patients with FSGS showed a decrease in the alpha-galactosidase A enzyme activity (which results in deposition of globotriaosylceramides in lysosomes) which correlated with sphingolipid deposition and podocyte injury leading to disease progression [75].

#### 3.1.3. Alport Syndrome

Alport syndrome is a rare inherited genetic disorder caused by mutations in collagen genes, especially in *COL4A3*, *COL4A4* and *COL4A5*, and is characterized by progressive kidney disease and abnormalities of the inner ear and eye. A study of a cohort of patients with Alport syndrome showed that the podocyte detachment rate, as measured by podocin mRNA presence in urine, was increased more than 11-fold when compared to healthy individuals [78]. The podocyte number per glomerulus in biopsies from patients with Alport syndrome was normal at the time of birth, but podocyte detachment increased leading to ESKD by 22 years [78] and the degree of proteinuria correlated with a reduction in the podocyte number, glomerulosclerosis and decreased renal function [78]. Only very limited information on the contribution of SLs to the progression of renal disease in Alport syndrome is currently available. Abnormalities in the levels of sulfatides were shown to cause diseases such as renal cell carcinoma and protein overload nephropathy [79,80]. Analysis of serum lipids by matrix-assisted laser desorption ionization time of flight mass spectrometry (MALDI-TOF MS) revealed that the level of serum sulfatides decreased with kidney dysfunction, but increased in kidneys and correlated with the high incidence of cardiovascular disease in patients with end-stage renal failure [79]. Recently, a drastic increase in sulfatide levels, especially of sulfohexosyl ceramides (d18:2/24:0 and d18:2/16:0), was described in kidneys of mice with Alport syndrome compared to wildtype controls [81], suggesting a role of SL in the development of renal disease in Alport syndrome. In the early stages of Alport syndrome, sulfatides are localized in renal tubules, suggesting the contribution of sulfatides to the development of tubulointerstitial fibrosis in Alport syndrome [81]. It is speculated that sulfatides contribute to disease progression by mediating the reabsorption of profibrotic chemokines and the stress-induced secretion of cytokines in Alport podocytes, thereby affecting monocyte infiltration into the renal interstitium and triggering the production of TGF-β1, which plays an important role in the pathogenesis of renal fibrosis [81]. More research is needed to understand the role of SLs and their contribution to the development and progression of Alport syndrome.

#### 3.1.4. IgA Nephropathy 

IgA nephropathy is characterized by an increase of immune complexes in mesangial cells resulting in mesangial proliferation and glomerular inflammation. Forty percent of patients with IgA nephropathy progress to ESKD [82]. In renal biopsies from patients with IgA nephropathy, mesangial hypercellularity and the accumulation of immune complexes in mesangial matrix can be observed [83]. Immunofluorescent studies showed that IgA deposits are detected mainly in mesangial cells. Electron microscopy revealed that IgA deposits can also be found in the capillary walls and occasionally in endothelial cells [83,84]. IgA deposits in the mesangial cells promote the production of cytokines that act as a pathogenic factor of podocyte injury in IgA nephropathy [85]. An increase in the number of podocytes in the urine is associated with IgA nephropathy progression [86]. Podocyte loss and foot process effacement in IgA nephropathy patients correlated with proteinuria [87]. In addition to proteinuria and hematuria, podocyte lesions were observed in patients with IgA nephropathy [88]. Microarray studies using mRNA isolated from kidneys of HIGA (a model for IgA nephropathy) mice demonstrated an increased renal expression of S1P receptors, suggesting the role of S1P receptor signaling in IgA nephropathy [89,90]. Similarly, increased S1P levels were found in urine and serum of patients with IgA nephropathy and S1P levels positively correlated with proteinuria [58,90]. The increase in S1P levels might be due to increased activity of SPHKs, suggesting that inhibition of the activity of SPHKs may lead to a decrease of S1P levels in the urine and plasma of patients with IgA nephropathy. Treatment of HIGA mice with JTE013, a S1PR2 antagonist, rescued them from proteinuria [90], indicating a major role for S1P in the development of IgA nephropathy [58,91], possibly mediated by S1PR2. It was demonstrated that S1P bound to APOM is protective against IgA nephropathy by suppressing the downstream signaling of S1PR1 and S1PR3, whereas S1P bound to albumin showed deteriorating effects through S1PR2 [90].

#### 3.1.5. Lupus Nephritis

Lupus nephritis (LN) is a kidney disease caused by systemic lupus erythematosus (SLE). SLE is a chronic autoimmune disease that can damage several organs including kidneys. About one half of patients with SLE develop kidney disease which can lead to ESKD [92]. While light microscopy studies revealed a membranoproliferative pattern of glomerular injury with extensive subendothelial deposits, resulting in wire loop appearance, electron microscopy studies showed mesangial and subendothelial deposits of IgG and tubuloreticular aggregates [93]. Proteinuria is one of the clinical manifestations of LN [94,95] and podocyte injury was shown to be one of the main causes of proteinuria in LN and correlates with the severity of the disease [96,97]. Nestin, a cytoskeleton protein which is stably expressed in podocytes, is associated with podocyte injury in LN [98], and was found to protect podocytes by regulating the expression and phosphorylation of Nephrin, thereby reducing proteinuria in mice with LN [98]. In another study, increased expression of Tim-1 protein protected podocytes in lupus-prone mice by inhibiting the IgG-induced inflammatory response in podocytes and reducing tumor necrosis factor α (TNF-α), interleukin (IL)-6 and IL-1β expression, and inducing autophagy [99]. A clinical study reported that dysregulation of circulating SLs (especially ceramides, which were increased whereas sphingoid bases were decreased) is associated with SLE [100]. LC-MS studies showed that sphingosine levels are significantly decreased in the serum of LN patients compared to healthy controls [101]. However, in another study, sphingosine levels were found increased in LN patients [102]. In other studies, it was demonstrated that ceramide (C16, C18, C20 and C24) levels are increased in the serum and plasma of LN patients [100,102]. Increase in C16 and C20 ceramide levels in plasma and serum significantly correlated with proteinuria in LN patients [102], hence blood ceramides were shown to be novel markers for renal impairment in SLE [102]. C16-Lactosylceramide levels were found elevated in the glomeruli as well as in the urine of LN patients when compared to healthy controls [103]. The increase in lactosylceramides is due to increased neuraminidases activity, especially neuraminidase 1 (NEU1, which is localized to lysosomes), and which removes sialic acids from gangliosides and proteins. Because the SREBP-1c pathway controls the transcription of neuraminidases, targeting the SREBP-1c pathway could represent a valid therapeutic strategy for the treatment of patients with LN.

#### 3.1.6. Fabry’s Disease

Fabry’s disease is characterized by deficient activity of α-galactosidase A, a lysosomal hydrolase, which leads to the accumulation of globotriaosylceramide (Gb3) in patients’ tissues. Though this condition leads to ESRD, the mechanism by which the damage occurs is poorly understood, especially the contribution of SLs to Fabry’s disease. Light microscopy of glomeruli demonstrated podocytes with foamy appearing vacuoles, mesangial widening and varying degrees of glomerular obsolescence. Electron microscopy studies reveal the presence of enlarged lysosomes packed with lamellated membrane structures [71].

Damage to the kidneys can be attributed to the accumulation of Gb3, especially in podocytes at an early age, leading to the detachment of podocytes [104,105] coinciding with dysregulated autophagy that contributes to podocyte damage in Fabry’s disease [106]. Loss of podocytes into the urine correlated with the severity of Fabry nephropathy [104]. Increases in the levels of Gb3 and lyso-Gb3 were observed in the serum and plasma of patients, which is a hallmark of Fabry’s disease [107,108]. Gb3 accumulation was associated with an increase in autophagosomes and a decrease in mTOR and AKT signaling leading to the podocyte injury in Fabry’s disease [106]. Hence, studies focusing on the connection between reduced activity of α-galactosidase A and dysregulated autophagy are likely to uncover the mechanism leading to kidney damage in Fabry’s disease.

#### 3.1.7. COVID-Mediated Kidney Injury

Severe acute respiratory syndrome coronavirus (SARS-CoV-2), now designated as coronavirus disease 2019 (COVID-19) has rapidly spread around the world. SARS-CoV2 interacts with human angiotensin converting enzyme II (ACE2) via its spike protein [109]. ACE2 is expressed in several tissues including human kidney and bladder [110] and SARS-CoV2 was shown to directly infect human kidney organoids via the ACE2 receptor [111]. Acute kidney injury (AKI) is observed in a subset of COVID-19 patients, suggesting that SARS-CoV2 infection might result in kidney injury [112] by directly infecting kidney tubules and inducing tubular damage [113]. It is believed that the virus enters the glomerular capillaries and infects glomerular endothelial cells and podocytes [114]. Collapsing glomerulopathy has been reported in COVID-19 patients [115]. Autopsy results from COVID-19 patients revealed glomerular fibrin thrombi with ischemic collapse and peritubular erythrocyte aggregation [116]. In a study comprised of six patients with COVID-19-related kidney disease, podocyte injury with foot process effacement and protein overload tubulopathy was observed, suggesting that podocyte damage occurs during COVID-19 infection [117]. SL profiling by mass spectrometry in the sera of COVID-19 patients showed a significant difference in the levels of SLs. In particular, dihydrosphingosine, sphingosine, long chain and very long chain ceramides except ceramide C24:0 levels were increased with disease severity whereas SM and S1P were decreased [118]. ASMase and SPT levels were increased in severe and critically ill patients. Critically ill patients were characterized by high levels of dihydrosphingosine and dihydroceramide [118]. These data indicate that changes in SL profile might be linked to COVID-19 disease severity and SLs, especially ceramides C16:0, C18:0, C24:1, and sphingosine can be considered as prognostic biomarkers [118]. 

#### 3.1.8. Radiation-Induced Kidney Injury

Tissue exposure to radiations as a treatment strategy for cancer can cause damage to kidneys and can manifest with both glomerular and tubulointerstitial damage with proteinuria and reduced glomerular filtration rate [119]. An increase in ceramide levels, a decrease in sphingosine, neutral ceramidase activity and S1P levels was observed in podocytes after irradiation, leading to podocyte injury due to loss of filopodia, remodeling of cortical actin and relocation of actin binding protein ezrin from the plasma membrane to cytosol [12]. Interestingly, S1P was shown to play a protective role in radiation-induced podocyte injury. Treatment of irradiated podocytes with exogenous S1P attenuated cytoskeletal remodeling and podocyte injury post radiation [12].

#### 3.1.9. Treatment Strategies

Podocytes are the crucial components of kidney function and hence, unraveling the pathological mechanisms that cause podocyte dysfunction is essential for the development of therapeutics for the treatment of patients with glomerular diseases. SMPDL3b is a GPI anchored lipid raft protein [120,121] that was shown to have a role in glomerular diseases [31,50,122]. SMPDL3B protein levels were found elevated in glomeruli of patients with DKD. Podocytes treated with sera of DKD patients showed an increase in the SMPDL3b protein expression and actin reorganization, while knockdown of SMPDL3B protected podocytes from apoptosis [48]. We also demonstrated that overexpression of SMPDL3b in podocytes can lead to impaired phosphorylation of AKT in a C1P-dependent manner, thereby promoting podocyte injury in DKD, while podocyte-specific deletion of SMPDL3b or exogenous C1P administration rescues podocyte loss and prevents proteinuria [29]. We furthermore showed that the expression of SMPDL3b which acts as a potential C1P phosphatase [29,30] is decreased in podocytes of patients with recurrence of FSGS after transplantation [121], while overexpression of SMPDL3b in podocytes prevents actin cytoskeleton disruption and apoptosis [121]. In the same study, we demonstrated that rituximab, an anti CD20 monoclonal antibody, binds to SMPDL3b, thereby stabilizing it and preventing cytoskeleton disruption and apoptosis in podocytes treated with the sera of patients with recurrent FSGS. In patients, rituximab was suggested as a treatment option to prevent the recurrence of proteinuria after transplantation in some patients with FSGS [121,123,124,125]. Rituximab was also shown to prevent Adriamycin-induced nephropathy in rats [122] and radiation therapy was found to cause decreases in SMPDL3b protein expression and podocyte damage [12]. Taken together, these studies imply that SMPDL3b is a potential therapeutic target for the treatment of kidney diseases, in particular of glomerular diseases. Whether compounds targeting SMPDL3b without affecting CD20 would result in a similar degree of renoprotection remains to be established. Similarly, C1P treatments could also be considered as an alternative for the treatment of glomerular diseases as suggested by our studies in experimental DKD. However, development of a formulation that would allow for both proper pharmacokinetic properties of active sphingolipids and proper delivery to kidney cells is yet to be established. 

Finally, S1P receptor signaling was found to play a major role in preserving proper function of all cells within the glomerular filtration barrier. Treatment of rat and mouse models of DKD with a non-selective S1PR agonist FTY720 or a S1PR1 agonist SEW2871 was shown to reduce proteinuria [56], while inhibition of S1PR2 by berberine prevented renal injury in diabetic rats [126], suggesting that targeting S1P receptors or S1P signaling can lead to the development of therapeutics to treat glomerular diseases. As acid ceramidase deficiency leads to podocytopathy and NS [77], an acid ceramidase inducer could also represent a novel therapeutic target to prevent glomerular diseases. The potential drug targets to treat different kidney diseases are summarized in Table 1. 

Another strategy to target SLs is enzyme replacement therapy (ERT). ERT targeting SLs is used to treat Fabry’s disease [129]. ERT provides the enzyme that is deficient or non-functional in the body of patients, thus managing the levels of deficient enzymes. ERT demonstrated a beneficial effect in reducing accumulation of glycosphingolipdis in lysosomes. ERT with α galactosidase A prevented the Gb3 accumulation in podocytes in Fabry’s disease patients [130]. Another study showed ERT with agalsidase β reduced the Gb3 accumulation in podocytes and microalbuminuria in Fabry’s disease patient’s [131], thus highlighting the importance of ERT in treating nephropathy in Fabry’s disease patients. In addition to the above strategies, there are several “omics” databases of kidney diseases which store and exchange research discoveries, thus helping in the identification of new mechanisms and drug targets. For example, the role of the JAK-STAT pathway in the pathogenesis of DKD was identified by comparing the expression profile of glomerular genes of DKD patients to experimental mouse models of DKD using these databases [132,133,134,135].

### 3.2. Role of Sphingolipids in Tubulointerstitial Fibrosis and Acute Kidney Injury

SLs have been shown to damage proximal tubular cells in many models of acute kidney injury (AKI), which include renal ischemia reperfusion injury (IRI), myoglobinuric AKI and cisplatin-induced AKI [127,136,137]. The amount of ceramide levels present was found to be proportional to the extent of injury to proximal tubular cells [127,138], indicating that ceramides play a pivotal role in tubular cell damage possibly by inducing mitochondrial damage through reactive oxygen species (ROS) production and DNA damage [9,128]. Sphingosine from which ceramides can be synthesized mediates acute proximal tubular injury [139]. Sphingosine-mediated tubular damage can be due to the inhibition of mitogen activated protein kinases (MAPKs) activity, ERK1 and ERK2 and activation of stress activated protein kinases (SAPKs), JNK and p38 [140,141], which is associated with apoptosis induced by a reduction in the expression of growth factors [142]. Mass spectrometry analysis of SLs revealed that long and very long chain ceramide levels were increased in myoglobinuric-, cisplatin- and IRI-induced AKI causing injury to cells [127,136,137] which can be attenuated by treating cells with fumonisin B1, a ceramide synthase inhibitor [143]. Additionally, inhibiting ceramide synthesis by myriocin, a SPT complex inhibitor, prevented the cisplatin-mediated increase in ceramide levels and proximal tubular cell death by inducing pores in the mitochondrial outer membrane, mitochondrial outer membrane permeabilization and mitochondrial fragmentation [144,145,146,147,148]. In support, the accumulation of ceramides due to increased sphingomyelinase activity was found to cause necrotic cell death of proximal tubular cells [149], while the conversion of ceramides to glycosphingolipids reduced cell death in cisplatin-induced AKI [137], indicating that ceramide accumulation plays a major role in proximal tubular cell injury.

The SPHK1 and S1P axis was shown to mediate the pathogenesis of tubular epithelial cells in DKD [50,150]. In *db*/*db* diabetic mice, it was shown that S1P-induced activation of Rho kinase-mediated fibrosis in renal tubular cells through S1PR2 disturbed the distribution of the adhesion molecule, E-cadherin, along the plasma membrane and of alpha-smooth muscle actin (α-SMA) in cytoplasm [151]. SPHK2 knockout but not SPHK1 knockout mice exhibit aggravated injury after IRI, suggesting a protective role of SPHK2 in IRI [152]. Contrary to this, another study showed that S1P synthesized by SPHK1 is important in the protection from IRI and that SPHK1 overexpression improves renal function via inducing the expression of heat shock protein 27 (HSP27) [153] that acts as an antioxidant and inhibits apoptosis [154]. Similarly, S1P and S1PR1 play a major role in renal injury after IRI as activation of S1P-dependent signaling and S1PR1 antagonism prevents isoflurane-mediated renal IRI and inflammation in both proximal tubular cells (HK2) and mice [155]. Knockout of SPHK2 reduced renal fibrosis after AKI induced in folic acid or unilateral ischemia reperfusion by inducing the expression of IFN-gamma responsive genes such as *Cxcl9* and *Cxcl10* [156]. Inhibiting S1P synthesis mediated by SPHK1 and SPHK2 was found to reduce tubulointerstitial renal inflammation and fibrosis in DKD and in human HK2 cells by reducing the expression of fibronectin, collagen IV and macrophage chemoattractant protein 1 (MCP1) [157]. From these studies, it can be concluded that the SPHK and S1P axis plays an important role in the protection of proximal tubular cells from renal IRI [158,159]. 

S1P receptors also play a major role in tubular cell damage. Loss of S1PR1 in proximal tubular cells exacerbates IRI [55], which in turn can be inhibited by increasing S1P synthesis, S1PR1 receptor agonism and SPHK gene delivery [55,155]. Increased apoptosis is observed in S1PR1 knockout mice treated with cisplatin and activation of S1PR1 by FTY720 was shown to rescue cisplatin-induced AKI by stabilizing the mitochondrial function by decreasing cytochrome c release and regulation of BCL-2 proteins [160]. SEW2871 also prevented apoptosis and attenuated IRI whereas VPC4416, a S1PR1 antagonist, aggravated IRI and inflammation associated with tubular injury induced by LPS [55,161,162,163]. Interestingly, S1PR2 antagonists or mice that lack S1PR2 are protected from renal IRI and SPHK antagonism or S1PR2 agonism exacerbates renal IRI [153,155], indicating that S1P signaling mediated by S1PR1 protects tubular cells, whereas S1P signaling mediated by S1PR2 causes tubular cell injury. Though proximal tubular cells are highly susceptible cells to injury during renal IRI, renal endothelial cell damage is also observed. Loss of S1PR1 in renal endothelial cells prevented the recovery from IRI and even aggravated renal injury, inflammation and renal fibrosis [159]. S1PR1 was shown to suppress endothelial activation of leukocyte adhesion molecule expression and inflammation [159]. 

Endothelial dysfunction is a major contributor for hypertension that can lead to cardiovascular and kidney diseases [164,165,166]. Plasma sortilin levels were increased in hypertensive patients with endothelial dysfunction along with an increase in ASMase activity, plasma S1P and soluble NADPH oxidase 2 (NOX2)-derived peptide [167]. Sortulin induces endothelial dysfunction of NOX2 activation, increasing oxidative stress and altering SL metabolism. Sortulin levels are associated with altered SL metabolism and oxidative stress in patients with arterial hypertension [167]. LC-MS analysis of sortilin-treated endothelial cells showed a decrease in ceramides (C16–C24) with an increase in S1P [167]. S1P was shown to play a significant role in hypertension-mediated kidney damage [167,168,169]. Sortilin-induced endothelial dysfunction was prevented by knockdown of either ASMase or SPHK1 [167]. Sortilin was found to induce vascular oxidative stress by promoting Rac1-mediated activation of NOX2 that was prevented by inhibition of S1PR3-mediated signaling [167]. Another study showed that Nogo-B (ER membrane protein and reticulon-4 family protein) regulates hypertension through endothelial nitric oxide synthase (eNOS) pathway [169]. Endothelial S1P-S1PR1 signaling is important for eNOS pathway activation [170]. LC-MS analysis of SLs in Nogo-B-deficient primary endothelial cells isolated from mouse lung showed an increase in sphingosine and S1P [169]. Nogo-B inhibits SPT expression thereby controlling the S1P production and its signaling mediated by S1PR1, thus restoring the endothelial dysfunction [169]. Recently, it was shown that expression of genes related to SL metabolism were altered in renal tissues of hypertensive rats when compared to normotensive rats [168]. Expression of S1PR1 was increased in both spontaneously hypertensive rats that are stroke resistant (SHSR) and spontaneously hypertensive rats that are stroke prone (SHSP) when compared to normotensive Wistar Kyoto rats [168]. SHSR rats were characterized by reduced expression of CerS2 mRNA in kidneys, whereas SHSP rats were characterized by reduced protein expression of SPTLC2 in kidneys [168].

Glycosphingolipid globotriaosylceramide (Gb3), which is expressed in renal proximal tubular epithelia, was shown to influence the reabsorption of proteins and albumin in proximal tubules. Inhibition of Gb3 synthesis blocks the reabsorption of proteins and protects from tubular damage [171]. Lyso-Gb3 which induces the epithelial–mesenchymal transition (EMT) response in HK2 cells in a TGF-β-dependent pathway increases the expression of extracellular matrix (ECM) components, thus contributing to disease progression, which can be prevented by inhibiting TGF-β receptor [172]. 

Taken together, studies in tubular cells support the idea that SLs and tubular cell-specific targeting of genes regulating the SL pathway may represent a therapeutic option to prevent tubular injury and to attenuate the effects of renal IRI. 

#### Treatment Strategies

There are several new therapeutic targets for AKI being investigated which are involved in pathways related to inflammation, fibrosis, oxidative stress and mitochondrial function. Effects of oxidative stress in AKI were shown to be mitigated by antioxidants such as alpha-lipoic acid (ALA), selenium and propofol [173,174,175]. S1P analogue FTY720 was shown to decrease inflammatory markers in renal tubules in mice treated with cisplatin by enhancing the mitochondrial functions [160]. Both FTY720 and SEW2871 have a protective role in IRI in mouse kidneys by inhibiting the expression of pro-inflammatory molecules and lymphocyte egress [55,163]. Alkaline phosphatase protects from inflammation-mediated renal injury by dephosphorylating lipopolysaccharide and adenosine triphosphate [176]. Activation of selective adenosine 2A receptors which are potent inhibitors of inflammation protects from kidney injury following IRI in rats [177]. Mesenchymal stem cell derived extracellular vesicles in an AKI model showed that exosomes move to the site of injury and play a protective role in AKI by secreting growth factors such as VEGF and IGF-1 [178,179]. Vitamin D prevents the onset as well as ameliorates AKI, suggesting the connection of vitamin D to AKI [180]. Sulfotransferase (SULT) inhibitors (an enzyme responsible for detoxification of xenobiotics) such as meclofenamate prevented kidney injury in a rat model of AKI [181]. More recently, nanotechnology-based drug delivery is also being used to treat AKI [182]. Though most of these therapies have yet to be studied or are in the process of completion in humans, the observations in animal models portray a strong platform for future investigation in humans with a scope to develop new therapeutics. The potential sphingolipid drug targets to treat tubular damage are summarized in Table 1.

## 4. Conclusions

The studies discussed in this review imply that SLs have a prominent role in kidney diseases originating from glomerular and tubular cell injury. Hence, targeting enzymes involved in SL metabolism, such as SMPDL3b, SPHKs, acid ceramidases and glycosphingolipid synthases, targeting SL metabolites such as ceramides, S1P, C1P, GM3 and Gb3, or targeting S1P receptors could be potential new avenues to develop new therapeutic agents for the treatment of patients with renal diseases. 

## Figures and Tables

**Figure 1 ijms-23-04244-f001:**
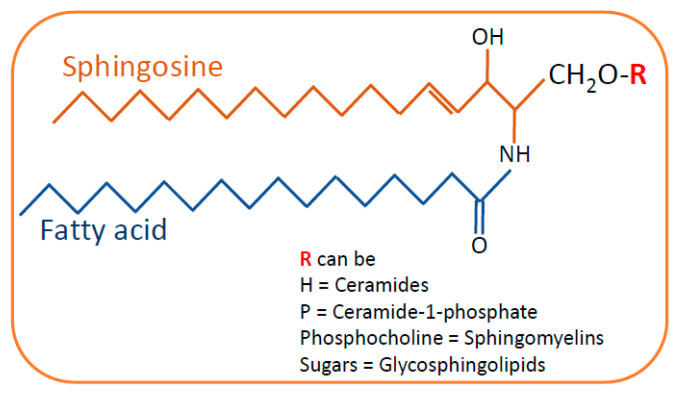
General structure of sphingolipids. Sphingosine is the backbone of the sphingolipid structure that is linked to fatty acids by an amide bond. Depending on the type of residue in the side chain, sphingolipids are categorized as ceramides, sphingomyelins and glycosphingolipids.

**Figure 2 ijms-23-04244-f002:**
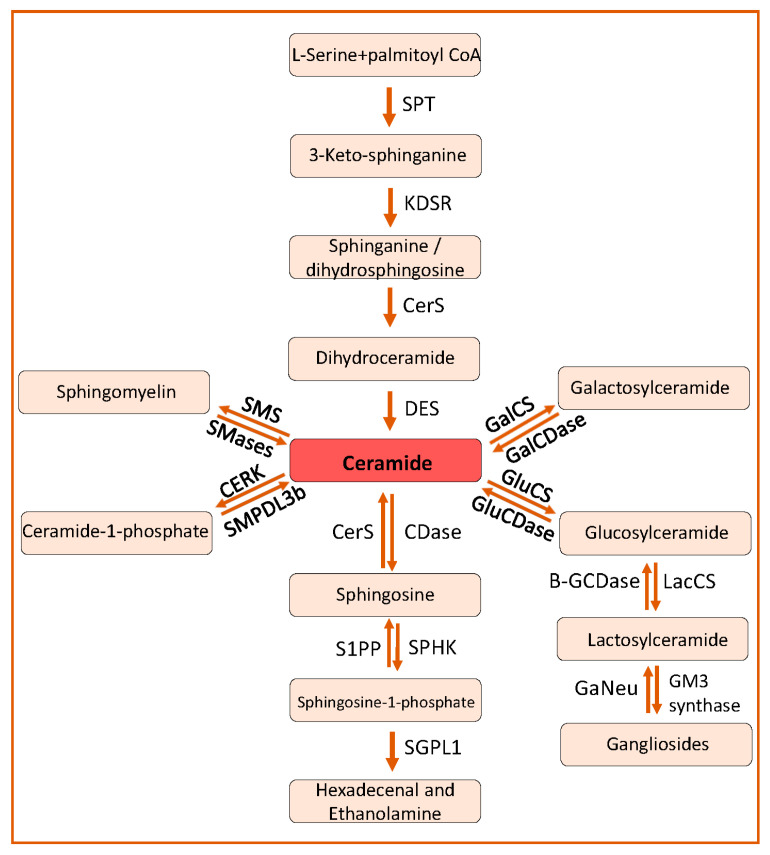
Sphingolipid pathway. The de novo sphingolipid metabolic pathway commences by the condensation of L-serine and palmitoyl-CoA to form 3-ketosphinganine, which ultimately is converted to ceramides by the action of KDSR, CerS and DES, respectively. Ceramides can also be synthesized from sphingomyelin by SMS, from glucosylceramide by GluCs and from galactosylceramide by GalCS. Ceramides can be phosphorylated to ceramide-1-phosphate by CERK and SMPDL3b can dephosphorylate ceramide-1-phosphate to ceramide. Ceramides can be deacylated to sphingosine by Cdases and sphingosine can be phosphorylated to form sphingosine-1-phosphate by SPHK. Sphingosine-1-phosphate can finally be broken down to hexadecanol and ethanolamine. SPT, serine palmitoyl transferase; KDSR, 3-ketosphinganine reductase; CerS, ceramide synthtase; DES, dihydroceramide desaturase; SMS, sphingomyelin synthetase, SMase, sphingomyelinase; CERK, ceramide kinase; SMPDL3b, Sphingomyelin Phosphodiesterase Acid Like 3b; CDase, ceramidase; SPHK, spingosine kinase; S1PP, spingosine-1-phosphate phosphatase; GluCS, glycosylceramide synthase; GalCS, galactosylceramide synthase; GluCDase, glycosylceramidase; GalCDase, galactosylceramidase; LacCs, lactosylceramide synthase; Β-GCDase, β-galactosidase; GaNeu, Ganglioside neuraminidase.

**Figure 3 ijms-23-04244-f003:**
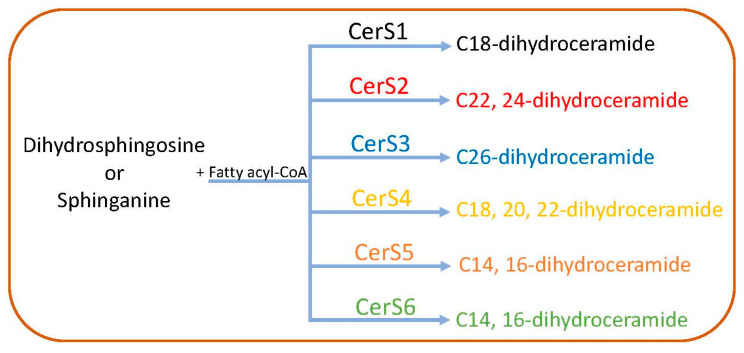
Specificity of ceramide synthases. There are six ceramide synthases that synthesize dihydroceramides with specific fatty acid chain lengths. Ceramides are formed from sphinganine or dihydrosphingosine by the attachment of acyl groups to the sphingosine backbone by ceramide synthases.

**Table 1 ijms-23-04244-t001:** Summary of sphingolipid species altered in different types of kidney diseases and the potential drug targets for the treatment of the different kidney diseases.

Kidney Disease	Sphingolipids Altered	Potential Drug Targets	References
Diabetic kidney disease	Sphingosine, Ceramides, S1P, C1P, Glycosphingolipids	CerS, ASAH1, SPHKs, S1PR1/S1PR2, SMPDL3b	[25,44,45,54,76,103,108]
Focal segmental glomerulosclerosis	Ceramides, S1P	ASAH1, SGPL1	[57,76,77]
Alport syndrome	Sulfohexosyl ceramides	CerS	[78]
IgA nephropathy	S1P	SPHKs, S1PR2	[56,82]
Lupus nephritis	Sphingosine, Ceramides, Lactosylceramides	CerS, NEU1	[86,88,89]
Fabrys disease	Gb3	α-galactosidase A	[90,91,92]
COVID mediated kidney injury	Sphingosine, Ceramides	SMase, SPT	[100]
Radiation induced kidney injury	S1P	SMPDL3b, SPHKs	[45]
Acute kidney injury	Sphingosine, Ceramides, S1P	CerS, SPHKs, S1PR1/S1PR2	[53,120,124,125,127,128]

Sphingosine-1-phosphate (S1P), Ceramide-1-phosphate (C1P), Globotriaosylceramide (Gb3), Ceramide synthases (CerS), Acid ceramidase (ASAH1), Sphingosine kinases (SPHKs), Sphingomyelin phosphodiesterase acid like 3b (SMPDL3b), Sphingosine-1-phosphate lyase (SGPL1), S1P receptors (S1PR), Neuraminidase 1 (NEU1), Sphingomyelinases (SMases), Serine palmitoyl transferase (SPT).

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
