# Peer review of "Implications of Sphingolipid Metabolites in Kidney Diseases"

_ijms, 2022, doi:10.3390/ijms23084244_

Round 1

Reviewer 1 Report

The review article by Mallela et al. gives a summary of the literature concerning sphingolipids and renal diseases. The review structure opens by introducing the reader to sphingolipid biology and biosynthesis, giving details of (intermediary) metabolites, relevant enzymes and their established functions. This followed by sections on various diseases of the kidney wherein the contributions of different sphingolipids are described. Finally, treatment strategies are discussed. Overall, the manuscript gives some insight into how sphingolipids contribute to different renal diseases and is a worthy topic to cover. However, there are improvements to be made.

It is unclear how the authors selected the renal diseases to highlight. Whilst glomerular syndrome, diabetic nephropathy, chronic kidney disease and more rare diseases are worthwhile to discuss, the sections on ischemia-reperfusion injury, acute kidney injury and fibrosis were kept rather small. There are also some glaring omissions, such as renal cancer, nephrotoxicity by toxic metals or drugs, or glomerulonephritis. Why were they not included? Possibly there is little to no literature concerning sphingolipids and these disease. A couple of explanatory sentences for the selection of renal diseases would be welcomed and additional other major renal diseases should be included.

Some of the diseases presented, e.g. IgA nephropathy, Lupus nephritis, Fabry’s, Covid-19 are so brief, questioning the value of one paragraph. Here, the pathology of the diseases could be expanded.

The section on sphingolipids and diseases (section 3) is completely superfluous in a review article on kidney diseases. Please omit or integrate into the manuscript with a kidney focus.

The section on treatment strategies would be better placed at the end of each section rather being combined in one large section at the end of the manuscript.

Some parts purely describe sphingolipid changes without addressing the impact of these changes. For example, on p.10, p.14-16 of the PDF file. Suggestions and speculations would make it more interesting and stimulating. For example, on p.9, it is stated that ceramides in the urine correlated with proteinuria. Here it would’ve been interesting to make the statement that ceramides are most likely protein-bound to albumin. Has this been reported in the blood? What is the impact on renal function? Protein endocytosis, etc?

Please indicate if mRNA or protein levels are altered as well as stating the method of sphingolipid analysis. This would aid the reader to interpret the reported data.

Figure 1 was missing.

The title should be edited to reflect the manuscript content. The alterations in sphingolipids has been detailed in diseases of the kidney rather than signaling pathways.

The sphingosine/S1P rheostat was not mentioned. Is there a reason for this?

Author Response

It is unclear how the authors selected the renal diseases to highlight. Whilst glomerular syndrome, diabetic nephropathy, chronic kidney disease and more rare diseases are worthwhile to discuss, the sections on ischemia-reperfusion injury, acute kidney injury and fibrosis were kept rather small. There are also some glaring omissions, such as renal cancer, nephrotoxicity by toxic metals or drugs, or glomerulonephritis. Why were they not included? Possibly there is little to no literature concerning sphingolipids and these disease. A couple of explanatory sentences for the selection of renal diseases would be welcomed and additional other major renal diseases should be included.

We thank the reviewer for the suggestions to include other renal diseases. Though there is enormous literature related to ischemia-reperfusion injury, acute kidney injury and fibrosis, less is known about the contribution of sphingolipids to these diseases. We have now implemented most of the studies investigating the role of sphingolipids in ischemia-reperfusion injury, acute kidney injury and fibrosis. We did not find any studies describing the role of sphingolipids in other renal diseases such as Renal Vasculitis, Acute Post Streptococcal Glomerulonephritis, Mixed Cryoglobulinemia, Membranoproliferative Glomerulonephritis, IgA Vasculitis, Membranous Nephropathy, Minimal Change Disease, nephrotoxicity by toxic metals and drugs, hence they were not included in the manuscript.

Some of the diseases presented, e.g. IgA nephropathy, Lupus nephritis, Fabry’s, Covid-19 are so brief, questioning the value of one paragraph. Here, the pathology of the diseases could be expanded.

We thank the reviewer for this input. There is very limited literature describing the connection between sphingolipids and IgA Nephropathy, Lupus Nephritis, Fabry’s, Covid-19. We have updated and expanded the sections as per the reviewer’s suggestion.

The section on sphingolipids and diseases (section 3) is completely superfluous in a review article on kidney diseases. Please omit or integrate into the manuscript with a kidney focus.

We thank the reviewer for this suggestion. We have omitted this section from the manuscript.

The section on treatment strategies would be better placed at the end of each section rather being combined in one large section at the end of the manuscript.

We thank the reviewer for this suggestion. We have placed the treatment strategies at the end of each section.

Some parts purely describe sphingolipid changes without addressing the impact of these changes. For example, on p.10, p.14-16 of the PDF file. Suggestions and speculations would make it more interesting and stimulating. For example, on p.9, it is stated that ceramides in the urine correlated with proteinuria. Here it would’ve been interesting to make the statement that ceramides are most likely protein-bound to albumin. Has this been reported in the blood? What is the impact on renal function? Protein endocytosis, etc?

We thank the reviewer for this comment. We have updated the manuscript as necessary.

Please indicate if mRNA or protein levels are altered as well as stating the method of sphingolipid analysis. This would aid the reader to interpret the reported data.

We thank the reviewer for this comment. The manuscript was updated accordingly.

Figure 1 was missing.

We apologize for the missing figure and have included the figure in the updated version of the manuscript.

The title should be edited to reflect the manuscript content. The alterations in sphingolipids has been detailed in diseases of the kidney rather than signaling pathways.

We thank the reviewer for this suggestion. We have implemented changed the title accordingly.

The sphingosine/S1P rheostat was not mentioned. Is there a reason for this?

We thank the reviewer for this observation. We have included the term of the sphingosine/S1P rheostat in the manuscript.

Reviewer 2 Report

Mallela and colleagues review the role of sphingolipids in kidney disease.  The review is balanced and comprehensive. This topic is timely and could be of broader interest.

1) A general introduction in sphingolipid structure would be appreciated. This reviewer could not find Figure 1. It will be also of interest to learn how strong various drug targets are expressed in the different cell types of the kidney -  as shown by reductionist or omics studies.

2) It appears that most of the models analyzed here are preclinical models (which is appropriate). It would be appreciated to add and separate clinical relevance where possible. It would be also of interest to clearly state where disturbances in sphingolipid metabolism have been observed (i.e. through omics studies).

3) It is not entirely clear for this reviewer how the different "types" of kidney disease were selected. It appears that some important ones are missing, i.e. kidney disease associated with hypertension.

4) This reviewer would appreciate a table that would give an overview of central points of the narrative review - that would summarize the central points and the status important for further drug development of this target.

Author Response

Mallela and colleagues review the role of sphingolipids in kidney disease.  The review is balanced and comprehensive. This topic is timely and could be of broader interest.

 1) A general introduction in sphingolipid structure would be appreciated. This reviewer could not find Figure 1. It will be also of interest to learn how strong various drug targets are expressed in the different cell types of the kidney -  as shown by reductionist or omics studies.

We thank the reviewer for the input. We have updated the manuscript with sphingolipid structure. Figure 1 is also updated in the manuscript. Few sentences were added on the identification of drug targets in kidney diseases.

2) It appears that most of the models analyzed here are preclinical models (which is appropriate). It would be appreciated to add and separate clinical relevance where possible. It would be also of interest to clearly state where disturbances in sphingolipid metabolism have been observed (i.e. through omics studies).

We thank the reviewer for the suggestions. We have updated the manuscript with information on relevant clinical studies.

3) It is not entirely clear for this reviewer how the different "types" of kidney disease were selected. It appears that some important ones are missing, i.e. kidney disease associated with hypertension.

We have updated the manuscript with some sentences mentioning why we choose different types of kidney diseases. Though there is a recent review mentioning that contribution of sphingolipids to hypertension mediated kidney damage needs to be investigated (Kidney360 March 2021, 2 (3) 534-541; DOI: https://doi.org/10.34067/KID.0006322020). We could not find any literature on how sphingolipids are implicated in the pathogenesis of hypertension related renal damage.

4) This reviewer would appreciate a table that would give an overview of central points of the narrative review - that would summarize the central points and the status important for further drug development of this target.

We thank the reviewer for the input. We have summarized the sphingolipid changes in kidney changes in the form of a table 1.
